# Differential Protein Expression Patterns of HOXA13 and HOXB13 Are Associated with Bladder Cancer Progression

**DOI:** 10.3390/diagnostics13162636

**Published:** 2023-08-09

**Authors:** Fee-Wai Chin, Huzlinda Hussin, De-Ming Chau, Teng-Aik Ong, Rosna Yunus, Azad Hassan Abdul Razack, Khatijah Yusoff, Soon-Choy Chan, Abhi Veerakumarasivam

**Affiliations:** 1Department of Biomedical Sciences, Faculty of Medicine and Health Sciences, Universiti Putra Malaysia (UPM), Serdang 43400, Selangor, Malaysia; feewaic@gmail.com (F.-W.C.); deming@upm.edu.my (D.-M.C.); 2Department of Pathology, Faculty of Medicine and Health Sciences, Universiti Putra Malaysia (UPM), Serdang 43400, Selangor, Malaysia; huzlinda@upm.edu.my; 3Department of Surgery, Faculty of Medicine, Universiti Malaya, Kuala Lumpur 50603, Malaysia; taong@um.edu.my (T.-A.O.); azadrazack@gmail.com (A.H.A.R.); 4Department of Pathology, Hospital Kuala Lumpur, Kuala Lumpur 50586, Malaysia; yrosna@yahoo.com.my; 5Department of Microbiology, Faculty of Biotechnology and Biomolecular Sciences, Universiti Putra Malaysia (UPM), Serdang 43400, Selangor, Malaysia; kyusoff@nibm.my; 6Malaysia Genome and Vaccine Institute, National Institutes of Biotechnology Malaysia, Kajang 43000, Selangor, Malaysia; 7School of Liberal Arts, Science and Technology, Perdana University, Kuala Lumpur 50490, Malaysia; 8School of Medical and Life Sciences, Sunway University, Bandar Sunway 47500, Selangor, Malaysia

**Keywords:** biomarker, bladder cancer, *HOXA13*, *HOXB13*, immunohistochemistry

## Abstract

Bladder cancer is a common urological cancer and has the highest recurrence rate of any cancer. The aim of our study was to profile and characterize the protein expression of homeobox A13 (*HOXA13*) and homeobox B13 (*HOXB13*) genes in Malaysian bladder cancer patients. The protein expression of HOXA13 and HOXB13 in formalin-fixed paraffin-embedded (FFPE) bladder cancer tissues was determined by immunohistochemistry (IHC) analysis. The association between HOXA13/HOXB13 protein expression and demographic/clinicopathological characteristics of the bladder cancer patients was determined by chi-square analysis. Approximately 63.6% of the bladder cancer tissues harbored high HOXA13 expression. High HOXA13 expression was significantly associated with non-muscle invasive bladder cancer, lower tumor grade, higher number of lymph node metastases, and recurrence risk. In contrast, low HOXB13 expression (including those with negative expression) was observed in 71.6% of the bladder cancer tissues analyzed. Low HOXB13 expression was significantly associated with muscle-invasive bladder cancer, higher tumor stage, tumor grade, and metastatic risk. Both HOXA13 and HOXB13 protein expression were found to be associated with bladder tumorigenesis. The putative oncogenic and tumor suppressive roles of *HOXA13* and *HOXB13*, respectively, suggest their potential utility as biomarkers in bladder cancer.

## 1. Introduction

Globally, more than half a million bladder cancer patients were diagnosed, and 212,536 deaths occurred in 2020 [1]. The predominant histologic type of bladder cancer is urothelial cell carcinoma (UCC), which accounts for more than 90% of the cases [2]. At diagnosis, approximately 70% of UCCs are non-muscle invasive bladder cancer (NMIBC; stages Ta–T1), while the remaining 30% are muscle invasive bladder cancer (MIBC; stages T2–T4) [3]. However, between 50 and 70% of NMIBC patients eventually develop tumor recurrence within five years of undergoing transurethral resection of the bladder tumor (TURBT). Of these, 10–20% of NMIBC patients will eventually progress to MIBC [4,5]. Due to the high risk of recurrence and disease progression, NMIBC patients require frequent and lifelong surveillance. Therefore, bladder cancer is associated with the highest medical cost per patient from first diagnosis to death as compared to other cancers [6,7].

Urine cytology and cystoscopy are routinely used for the diagnosis of bladder cancer. In addition, cystoscopy is also commonly used in surveillance follow-up. Urine cytology involves the microscopic examination of exfoliated cells/tumor cells from the voided urine of patients for the detection of bladder cancer. However, this test has low sensitivity for detecting low-grade tumors [8]. Cystoscopy is an invasive procedure that often causes significant discomfort and pain for patients. As a result, bladder cancer patients often have moderate to low compliance with surveillance follow-up [9,10]. Hence, the development of effective biomarkers is crucial so that cystoscopy is used only for confirmation upon the detection of a specific and sensitive biomarker(s) in the urine.

Bladder cancer is characterized by increasing genomic instability and the cohesive dysregulation of the transcriptome that influence cellular and clinical phenotypes, resulting in the identification of candidate genes that harbor significant biomarker potential [11,12,13]. The human homeobox gene family is a group of regulatory genes that encode embryogenesis-associated transcription factors. These genes include homeobox A13 (*HOXA13*) and homeobox B13 (*HOXB13*) [14,15]. *HOXA13* is involved in the formation and development of the limbs, reproductive system, and urinary tract [16]. *HOXB13* plays a crucial role in the development and maintenance of the skin [17].

Several studies have evaluated the utility of *HOXA13* as a biomarker to detect bladder cancer using voided urine samples [18,19]. The detection of *HOXA13* and *CDC2* mRNA in voided urine had an accuracy of 80% in determining whether a particular UCC was of high stage or grade. Moreover, *HOXA13* mRNA expression alone was able to detect low-stage or low-grade tumors with more than 90% specificity [20]. Their findings were supported by other studies [18,19] in which *HOXA13* expression was elevated in low-stage tumors (pTa–pT1) of bladder cancer. In addition, a combination of *HOXA13* and *BLCA-4* mRNA expression had 80% sensitivity and specificity in distinguishing between low- and high-grade tumors [18,19]. Hence, *HOXA13* has huge potential as a biomarker for the diagnosis and prognosis of bladder cancer.

*HOXA13* overexpression has been reported in bladder, brain, esophageal, gastric, liver, and prostate cancers [21,22,23,24,25,26,27]. The overexpression of *HOXA13* has been shown to promote tumor cell proliferation, migration, invasion, metastasis, and epithelial–mesenchymal transition, as well as inhibit apoptosis. Thus, *HOXA13* may play a role as an oncogene in these cancers. In contrast, *HOXB13* overexpression has been reported in liver, ovarian, and pancreatic cancers [28,29,30]. The overexpression of *HOXB13* was shown to promote tumor cell proliferation and inhibit apoptosis, suggesting an oncogenic role for *HOXB13* in these cancers. Nevertheless, some studies have reported that *HOXB13* is downregulated in colorectal, kidney, and prostate cancers [31,32,33]. Induced expression of *HOXB13* inhibited tumor cell proliferation and induced apoptosis, suggesting that *HOXB13* acts as a tumor suppressor gene in these types of cancers. In this study, both *HOXA13* and *HOXB13* were selected after conducting a comprehensive review of the existing literature on the dysregulation of homeobox genes in bladder cancer.

Although previous studies have shown the association of *HOXA13* and *HOXB13* with bladder cancer as well as the potential utility of *HOXA13* as a biomarker for bladder cancer [18,19,20], findings from those studies are far from conclusive and require further robust validation. As such, the role of *HOXA13* and *HOXB13* in bladder cancer is not well studied. Moreover, both genes have not been characterized in Malaysian bladder cancer patients. Therefore, our study aimed at profiling the protein expression of HOXA13 and HOXB13 in a cohort of formalin-fixed paraffin-embedded (FFPE) bladder cancer tissues from Malaysian bladder cancer patients. In addition, our study also aimed to determine the association of HOXA13/HOXB13 protein expression with various demographic and clinicopathological characteristics of bladder cancer patients.

## 2. Materials and Methods

### 2.1. Patients and Tissue Samples

Bladder cancer patients admitted to Hospital Kuala Lumpur, Malaysia, from 2002–2013 were involved in this study. Ethical approval was obtained from the Medical Research and Ethics Committee, Ministry of Health Malaysia (approval number NMRR-12-970-13096). In addition, written informed consent was obtained from the patients. Retrieval of 573 bladder cancer patients’ medical records was conducted from April 2013 to August 2016. The collected information includes patients’ demographic (gender, ethnicity, and age) and clinicopathological characteristics (tumor invasiveness, tumor stage, tumor grade, lymph node metastasis, metastasis, and tumor recurrence).

The FFPE bladder cancer tissue blocks of 164 patients were retrospectively collected from the Department of Pathology, Hospital Kuala Lumpur, and stored at room temperature. Histological examination was conducted to select tissue blocks with at least 80% tumor cellularity, whereas tissue blocks with large areas of tumor necrosis or an insufficient amount of tissue were excluded. A total of 110 bladder cancer patients were selected for this study, in which their tissue blocks were obtained from cystoprostatectomy, cystectomy, TURBT, cystectomy plus total abdominal hysterectomy, bilateral salpingo-oophorectomy (TAHBSO), and biopsy.

### 2.2. Tissue Sectioning

Sectioning of FFPE tissues was performed using a LEICA RM2135 microtome (Leica Biosystems, Wetzlar, Germany). The tissue blocks were sectioned into 4 µm thick sections. The tissue sections were placed in a Leica HI1210 water bath (Leica Biosystems, Germany) set at 42 °C and then fixed onto SuperFrost Plus^®^ glass slides (Thermo Fisher Scientific, Waltham, MA, USA). The slides were dried overnight at room temperature and then stored at 4 °C until further analysis.

### 2.3. Hematoxylin and Eosin (H&E) Staining

H&E staining was carried out using a Tissue-Tek Prisma Automated Slide Stainer (Sakura Finetek, Torrance, CA, USA). The H&E-stained slides were examined by a pathologist to ensure that the selected tissue blocks contained sufficient tumor cellularity for immunohistochemistry (IHC) analysis.

### 2.4. IHC Analysis

A total of 110 and 102 FFPE bladder cancer tissues were used for the IHC analysis of HOXA13 and HOXB13, respectively. There was insufficient tissue material in eight tissue blocks after the completion of the HOXA13 IHC analysis. Hence, HOXB13 IHC analysis was not conducted on these samples. IHC analysis was performed using the Dako REAL^TM^ Envision^TM^ Detection System, Peroxidase/DAB+, Rabbit/Mouse kit (Agilent Technologies, Santa Clara, CA, USA). FFPE human skin tissue was used as the positive control for HOXA13 IHC analysis, while FFPE human prostate cancer tissue was used as the positive control for HOXB13 IHC analysis. In addition, matched tissues stained without the primary antibody were used as negative controls. In every batch of IHC analysis, FFPE bladder cancer tissues were analyzed together with a positive and a negative control.

Tissue slides were deparaffinized by incubating at 60 °C for 1 h in an oven (Venticell, Munich, Germany), followed by immersing them twice in xylene for 5 min. The slides were rehydrated by immersing them in a series of decreasing concentrations of ethanol: 100%, 100%, 95%, 80%, and 70% for 5 min each. The slides were then rinsed under slow-running tap water for 5 min. Antigen retrieval was performed using a microwave oven (National, Kadoma, Japan), where the slides were immersed in 1× Tris-EDTA buffer (pH 9) and heated at high mode for 5 min until boiling, followed by defrosting mode for 10 min. The slides were then cooled at room temperature for 35 min. Next, a hydrophobic circle was drawn around the tissue section onto the glass slides using a Dako Pen (Agilent Technologies, Santa Clara, CA, USA). The slides were then washed thrice by immersing them in 1× TBS with 0.05% Tween 20 (TBST) buffer for 5 min each time. The slides were then incubated with 200 µL of Dako REAL^TM^ Peroxidase-Blocking Solution (Agilent Technologies, Santa Clara, CA, USA) for 10 min. The slides were then washed thrice with 1× TBST buffer for 5 min each time.

For HOXA13 IHC analysis, 200 µL of the rabbit polyclonal anti-HOXA13 primary antibody (ab106503; Abcam, Cambridge, UK) at an optimal 1:400 dilution (working concentration of 2.5 µg/mL) was added onto each slide and incubated at room temperature for 30 min. For HOXB13 IHC analysis, 200 µL of the rabbit monoclonal anti-HOXB13 primary antibody (ab201682; Abcam, Cambridge, UK) at an optimal 1:200 dilution (working concentration of 1.725 µg/mL) was added onto each slide and incubated at room temperature for 1 h. The same optimal dilution of the primary antibody was used for all the tissue samples analyzed. A negative control was prepared, whereby the primary antibody was replaced with Dako Antibody Diluent (Agilent Technologies, Santa Clara, CA, USA). The slides were then washed thrice with 1× TBST buffer for 5 min each time. Subsequently, 200 µL of the Dako REAL^TM^ Envision^TM^/HRP, Rabbit/Mouse (Agilent Technologies, Santa Clara, CA, USA), which is the secondary antibody, was added to each slide and incubated at room temperature for 30 min. The slides were then washed with 1× TBST buffer for 5 min each time.

Next, 200 µL of Dako REAL^TM^ DAB+ Chromogen (Agilent Technologies, Santa Clara, CA, USA), which was diluted with Dako REAL^TM^ Substrate Buffer (Agilent Technologies, Santa Clara, CA, USA) at 1:50 dilution, was added to each slide and incubated for 10 min. The slides were rinsed twice under slow-running tap water for 5 min each time. After that, the slides were counterstained with Dako Mayer’s Hematoxylin (Agilent Technologies, Santa Clara, CA, USA) at a 1:2 dilution for 30 s. The slides were then rinsed twice under slow-running tap water for 5 min each time. After the slides had been stained with the primary and secondary antibodies and counterstained, they were dehydrated by immersing them in a series of increasing concentrations of ethanol: 70%, 80%, 95%, 100%, and 100% for 3 min each time. The slides were then immersed in xylene twice for 3 min each time. Finally, the slides were mounted with DePex mounting medium (VWR, Lutterworth, UK), covered with glass coverslips (Marienfeld, Lauda-Konigshofen, Germany), and dried overnight at room temperature.

### 2.5. Scoring of Immunohistochemical Staining

The IHC-stained slides were examined under a light microscope (Olympus, Center Valley, PA, USA) and scored by a pathologist who was blinded to the patients’ demographic and clinicopathological characteristics. The slides were scored based on the percentage of tumor cells expressing positive staining and the intensity of staining. The final IHC score was calculated by multiplying the percentage of positive cells by the staining intensity. The IHC scoring system for HOXA13 was based on Hu et al. [24], as shown in Table 1. Low HOXA13 expression was determined for tissues with final IHC scores of 0, 1, 2, and 3, while high HOXA13 expression was determined for tissues with final IHC scores of 4, 6, and 9.

Meanwhile, the IHC scoring system for HOXB13 was based on Kristiansen et al. [34], as shown in Table 2. Low HOXB13 expression was determined for tissues with final IHC scores of 0, 1, 2, 3, and 4, while tissues with final IHC scores of 6, 8, 9, and 12 were determined to have high HOXB13 expression.

### 2.6. Statistical Analysis

Statistical analysis was performed using the IBM SPSS Statistics version 20 software (SPSS Inc., Chicago, IL, USA). The association between the expression of both proteins and the demographic or clinicopathological characteristics of the bladder cancer patients was determined using chi-square analysis. A *p*-value of less than 0.05 was considered statistically significant. Patients were excluded from the chi-square analysis if missing data were present for a particular clinicopathological characteristic that was being analyzed.

## 3. Results

### 3.1. Protein Expression of HOXA13 and HOXB13 in Bladder Cancer Tissues

The majority of the FFPE bladder cancer tissues (94.5%, 104/110) positively expressed HOXA13, while only 5.5% (6/110) of these tissues were negative for HOXA13. The HOXA13 protein was predominantly localized in the nucleus of the bladder cancer cells, but in some tissues, it was also localized in both the nucleus and cytoplasm (Figure 1).

Both H&E and IHC staining of HOXA13 are shown in Figure 2. For HOXA13 IHC analysis, 40.0% (44/110) of the tissues showed strong staining, 31.8% (35/110) showed moderate staining, and 22.7% (25/110) showed weak staining (Figure 2). It was found that 63.6% (70/110) of the tissues had high HOXA13 expression, while 36.4% (40/110) had low HOXA13 expression (including those with negative expression).

Positive HOXB13 expression was found in 57.8% (59/102) of the FFPE bladder cancer tissues. The remaining tissues (42.2%, 43/102) did not express HOXB13; there was no observable staining. When expressed, the HOXB13 protein was mainly localized in the nucleus of the bladder cancer cells, followed by nucleus and cytoplasm co-localization. In a small fraction of tissues, expression was limited to the cytoplasm only (Figure 3).

Both H&E and IHC staining of HOXB13 are shown in Figure 4. For HOXB13 IHC analysis, 24.5% (25/102) showed weak staining, 21.6% (22/102) showed moderate staining, and 11.8% (12/102) showed strong staining (Figure 4). Most of the tissues (71.6%, 73/102) had low HOXB13 expression (including those with negative expression), whereas the remaining (28.4%, 29/102) had high HOXB13 expression.

### 3.2. Protein Expression of HOXA13 and HOXB13 across Different Stages and Grades of Bladder Cancer

HOXA13 protein expression in 110 FFPE bladder cancer tissues across different tumor stages and grades is depicted in Figure 5A,B, respectively. High HOXA13 expression was observed in 78.3% (18/23) of pTa tumors, 73.1% (19/26) of pT1 tumors, 66.7% (14/21) of pT3 tumors, and 58.8% (10/17) of pT4 tumors, as compared to 39.1% (9/23) of pT2 tumors. Correspondingly, high HOXA13 expression was also observed in 85.0% (17/20) of grade 1 tumors and 67.2% (43/64) of grade 3 tumors, as compared to 38.5% (10/26) of grade 2 tumors.

HOXB13 protein expression in 102 FFPE bladder cancer tissues across different tumor stages and grades is depicted in Figure 6A,B, respectively. Low HOXB13 expression was observed in 87.0% (20/23) pT2 tumors, 94.4% (17/18) pT3 tumors, and 93.3% (14/15) pT4 tumors as compared to 42.9% (9/21) pTa tumors and 52% (13/25) pT1 tumors. Correspondingly, low HOXB13 expression was also observed in 68% (17/25) grade 2 tumors, and 83.1% (49/59) grade 3 tumors, as compared to 38.9% (7/18) grade 1 tumors.

### 3.3. Association of HOXA13 and HOXB13 Protein Expression with Demographic and Clinicopathological Characteristics of Bladder Cancer Patients

The association between HOXA13 protein expression and various patient demographics as well as clinicopathological characteristics is summarized in Table 3. HOXA13 protein expression was significantly associated with tumor invasiveness (*p* = 0.020), tumor grade (*p* = 0.003), lymph node metastasis (*p* = 0.027), and recurrence (*p* = 0.043). Based on the strength of the association, a strong association was found for tumor grade, while a moderate association was found for tumor invasiveness and lymph node metastasis. However, a weak association was found for recurrence. Notably, high HOXA13 expression was significantly associated with NMIBC, lower tumor grade, higher number of lymph node metastases, and recurrence risk (Table 3). In addition, HOXA13 protein localization was significantly associated with age at diagnosis (*p* = 0.001) and recurrence (*p* = 0.023) (Table 4).

Meanwhile, the association between HOXB13 protein expression and various patient demographics as well as clinicopathological characteristics is summarized in Table 5. HOXB13 protein expression was significantly associated with tumor invasiveness (*p* ≤ 0.001), tumor stage (*p* ≤ 0.001), tumor grade (*p* = 0.001), and metastasis (*p* = 0.005). There was a strong association for tumor invasiveness, tumor stage, and tumor grade, while there was a moderate association for metastasis. Notably, low HOXB13 expression was significantly associated with MIBC, higher tumor stage, higher tumor grade, and metastatic risk (Table 5). In addition, HOXB13 protein localization was significantly associated with tumor invasiveness *(p* ≤ 0.001), tumor stage (*p* ≤ 0.001), tumor grade (*p* = 0.032), and metastasis (*p* = 0.001). Moreover, nuclear HOXB13 expression was significantly associated with NMIBC, as a high frequency of nuclear localization was observed in NMIBC, indicating nuclear HOXB13 expression is able to distinguish NMIBC from MIBC (Table 6).

## 4. Discussion

Our study demonstrates that the HOXA13 protein is heterogeneously expressed in bladder cancer tissues with a predominantly nuclear localization. This finding is in contrast to that of Hu et al. [24], who reported that HOXA13 was mainly localized in the cytoplasm of bladder cancer cells. The cytoplasmic localization of HOXA13 might be due to the modulation of nuclear localization signals, the nuclear export sequence, and its interaction with cytoplasmic anchoring factors [35,36]. It is worth noting that the HOXA13 protein is a transcription factor that is synthesized in the endoplasmic reticulum, which is located in the cytoplasm. However, most transcription factors contain nuclear localization signals that mediate the transport of transcription factors into the nucleus to regulate gene transcription [37]. This explains the observation in our study that the HOXA13 protein is mainly localized in the nucleus. A similar finding was reported by Cantile et al. [38], in which their study found that HOXA13 was predominantly expressed in the nucleus of thyroid cancer cells rather than the cytoplasm.

Some of the findings of our study are similar to those of Hu et al. [24], in which high HOXA13 protein expression in bladder cancer was significantly associated with positive lymph node metastasis. High HOXA13 expression was more prevalent in positive lymph node metastasis, whereas low HOXA13 expression was more frequently observed in negative lymph node metastasis. In addition, our study found a strong association between HOXA13 protein expression and tumor grade. A meta-analysis study on various human cancers (bladder, esophageal, gastric, liver, oral, ovarian, pancreatic, and prostate cancers) that was conducted by Wen et al. [39] also showed that high HOXA13 expression was significantly associated with positive lymph node metastasis. In addition, high HOXA13 expression was also significantly associated with higher tumor grade and TNM stage as well as poor overall survival.

Interestingly, our study revealed a pattern of high HOXA13 expression in 73–78% of pTa and pT1 tumors and 58–67% in pT3 and pT4 tumors, but only in 39% of pT2 tumors. Correspondingly, high HOXA13 expression was found in 67–85% of grade 1 and grade 3 tumors but only 39% of grade 2 tumors. Our study partially contradicts that of Hu et al. [24], who reported that high HOXA13 expression was associated with high stage (pT2–pT4) and high grade (grades 2 and 3) bladder cancer. The discrepancy between the findings of these studies may be due to population heterogeneity, different types of primary antibodies used, and variations in the IHC protocol [40].

Our study suggests that *HOXA13* may be involved in different stages of bladder cancer progression. The roles of *HOXA13* in the differentiation and morphogenesis of the genitourinary tracts [41] may explain the higher HOXA13 expression in the well-differentiated pTa and pT1 tumors. When bladder cancer cells progress to a more advanced stage, the HOXA13 protein is overexpressed and potentially plays a different role. It has been suggested that the HOXA13 protein may be involved in the morphogenesis of cancer cells by causing abnormalities in cell adhesion, which affect the interactions between cancer cells and their extracellular matrix [42], thus enabling tumor invasion and metastasis. Taken together, *HOXA13* potentially plays different roles at the early and late stages of bladder cancer progression. These findings are significant, and further studies are needed to elucidate the precise stage-dependent role of *HOXA13* in bladder cancer.

Findings from our study suggest that *HOXA13* is a putative oncogene in bladder cancer and that it has the potential to be developed into a clinically applicable biomarker for the detection of bladder cancer. Thus, *HOXA13* can potentially complement the use of cystoscopy, whereby cystoscopy can be used for confirmation upon the detection of high HOXA13 expression. After collecting voided urine samples from bladder cancer patients, RNA or protein can be extracted from the exfoliated tumor cells that are present in the urine. The expression of *HOXA13* can be measured either at RNA or protein levels using RT-qPCR or ELISA, respectively. The use of *HOXA13* as a potential clinically applicable biomarker will help to avoid the need to impose mandatory cystoscopies, which often inhibit patients from coming and getting treatment or screening for recurrence. Perhaps the likelihood of compliance with surveillance follow-up will be higher if the *HOXA13* test is performed first instead of a cystoscopy.

Our study demonstrated that positive HOXB13 expression was found in 57.8% of the tissues analyzed, whereas negative HOXB13 expression was found in 42.2%, with the majority of the tissues being MIBC. In addition, low HOXB13 expression was significantly associated with MIBC, higher tumor stage (pT2–pT4) and grade (grades 2 and 3), and metastatic risk. These findings corroborate with those studies that were conducted on gastric and kidney cancers [33,43]. *HOXB13* expression was significantly lower in gastric cancer patients with lymph node metastasis, higher stage, and poorly differentiated (grade 3) tumors [43]. In addition, *HOXB13* methylation (loss of expression) in renal cell carcinoma was positively correlated with tumor grade and invasion [33].

Findings from our study showed that the HOXB13 protein was heterogeneously expressed in bladder cancer tissues and was mainly localized in the nucleus. Nuclear HOXB13 expression was significantly associated with NMIBC, allowing it to distinguish NMIBC from MIBC. Notably, the loss of HOXB13 protein expression was associated with progression to MIBC, suggesting *HOXB13* may have a tumor suppressive role in bladder cancer. Incidentally, the putative role of *HOXB13* as a tumor suppressor gene has been reported in other cancers, including colorectal [31], gastric [43], kidney [33], melanoma [44], and prostate cancer [45]. In contrast, Marra et al. [46] reported that HOXB13 is mainly localized in the cytoplasm of bladder cancer cells. Their study found that cytoplasmic HOXB13 protein expression was significantly higher in MIBC compared to NMIBC, indicating *HOXB13* promotes bladder cancer invasiveness. Thus, further investigation is needed to clarify the contradictory role of *HOXB13* in bladder cancer.

Our study demonstrated the association between both HOXA13 and HOXB13 expression with bladder tumorigenesis. However, only 110 patients fulfilled the inclusion criteria for this study, which is considered a small sample size for biomarker studies involving retrospectively collected specimens [47]. Small sample size studies may increase the chance of making inaccurate conclusions [48]. Hence, the findings of this study should be further validated by future studies with a larger sample size. This is crucial to thoroughly evaluating the potential of both *HOXA13* and *HOXB13* as biomarkers for bladder cancer.

## 5. Conclusions

Our study supports the oncogenic role of *HOXA13* in bladder cancer, as this gene is involved in the early and later stages of bladder tumorigenesis, while *HOXB13* may play a tumor suppressive role in bladder cancer. Thus, both the *HOXA13* and *HOXB13* genes have biomarker potential. Future work will focus on validating the clinical significance of *HOXA13* and *HOXB13* as biomarkers for bladder cancer. Moreover, it is necessary to conduct functional mechanistic candidate-gene approaches to clarify the emerging role of both genes in bladder cancer.

## Figures and Tables

**Figure 1 diagnostics-13-02636-f001:**
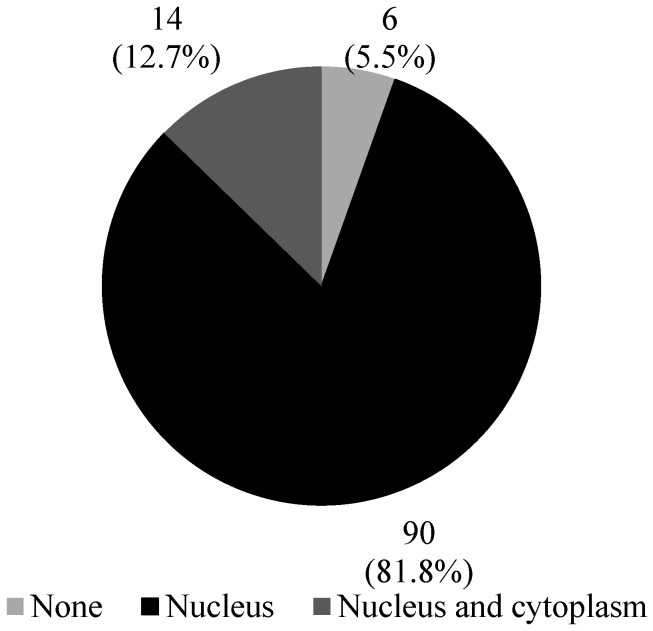
HOXA13 protein localization in FFPE bladder cancer tissues. HOXA13 protein was predominantly localized in the nucleus. In a small proportion of tissues, HOXA13 protein was also localized in both the nucleus and cytoplasm.

**Figure 2 diagnostics-13-02636-f002:**
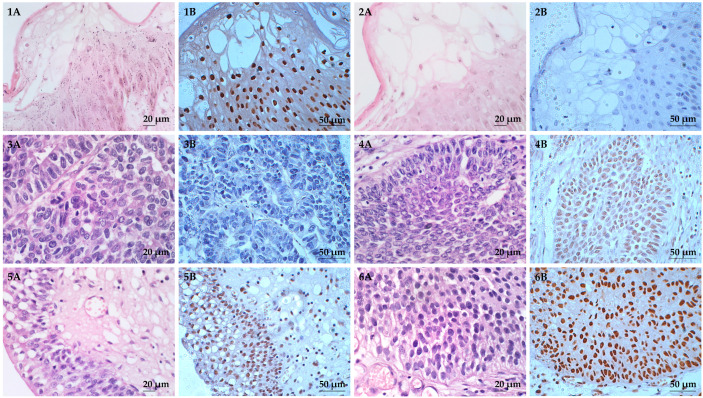
Hematoxylin and eosin (H&E) and immunohistochemical (IHC) staining of HOXA13 in FFPE bladder cancer tissues with nuclear localization. (**1A**–**6A**) are H&E staining, where (**1A**,**2A**) are FFPE human skin tissues and (**3A**–**6A**) are FFPE bladder cancer tissues. (**1B**–**6B**) are IHC staining of HOXA13. FFPE human skin tissues were used as (**1B**) positive and (**2B**) negative controls. FFPE bladder cancer tissues showed (**3B**) no staining, (**4B**) weak staining, (**5B**) moderate staining, and (**6B**) strong staining. All images were taken at 400× magnification.

**Figure 3 diagnostics-13-02636-f003:**
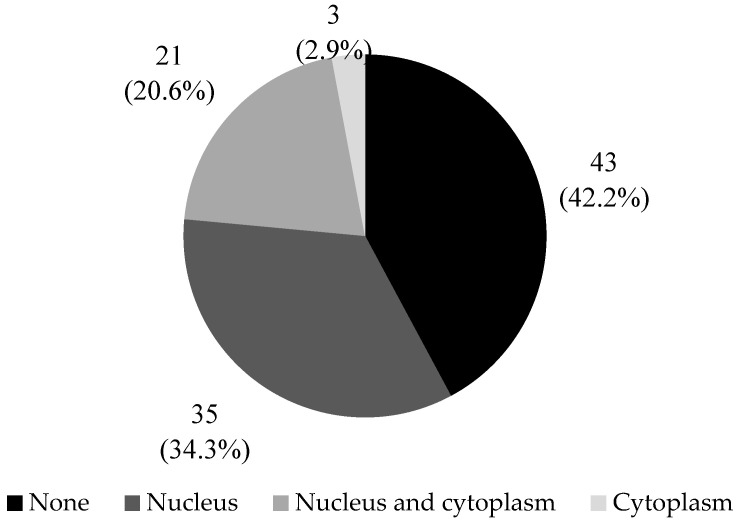
HOXB13 protein localization in FFPE bladder cancer tissues. HOXB13 protein was predominantly localized in the nucleus. In a smaller proportion of tissues, HOXB13 protein was localized in either both the nucleus and cytoplasm, or the cytoplasm only.

**Figure 4 diagnostics-13-02636-f004:**
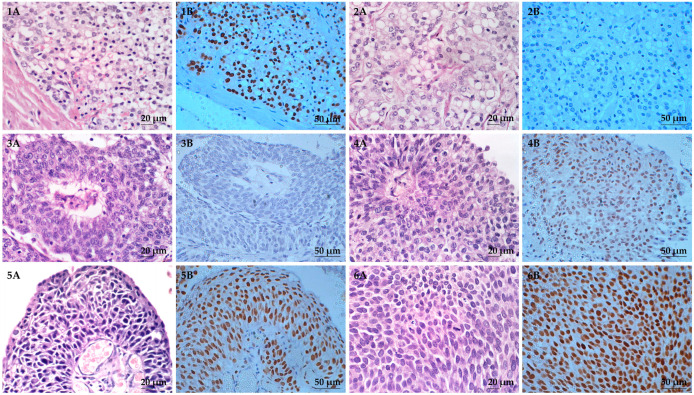
Hematoxylin and eosin (H&E) and immunohistochemical (IHC) staining of HOXB13 in FFPE bladder cancer tissues with nuclear localization. (**1A**–**6A**) are H&E staining where (**1A**,**2A**) are FFPE human prostate cancer tissues and (**3A**–**6A**) are FFPE bladder cancer tissues. (**1B**–**6B**) are IHC staining of HOXB13. FFPE human prostate cancer tissues were used as (**1B**) positive and (**2B**) negative controls. FFPE bladder cancer tissues showed (**3B**) no staining, (**4B**) weak staining, (**5B**) moderate staining, and (**6B**) strong staining. All images were taken at 400× magnification.

**Figure 5 diagnostics-13-02636-f005:**
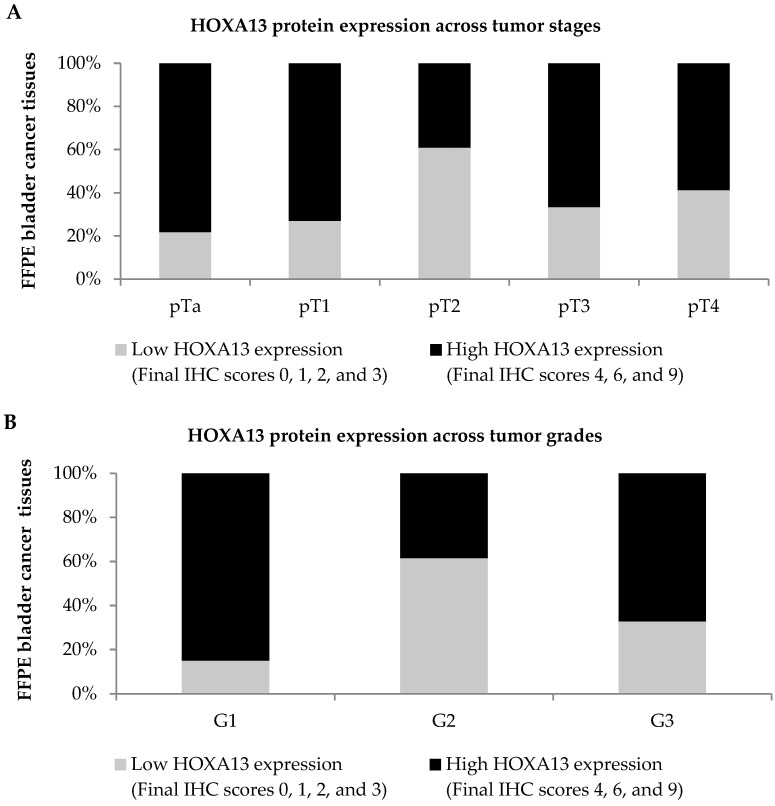
Protein expression of HOXA13 in FFPE bladder cancer tissues. The distribution of HOXA13 protein expression was stratified according to different (**A**) tumor stages and (**B**) tumor grades. HOXA13 protein expression was categorized into low and high HOXA13 expression.

**Figure 6 diagnostics-13-02636-f006:**
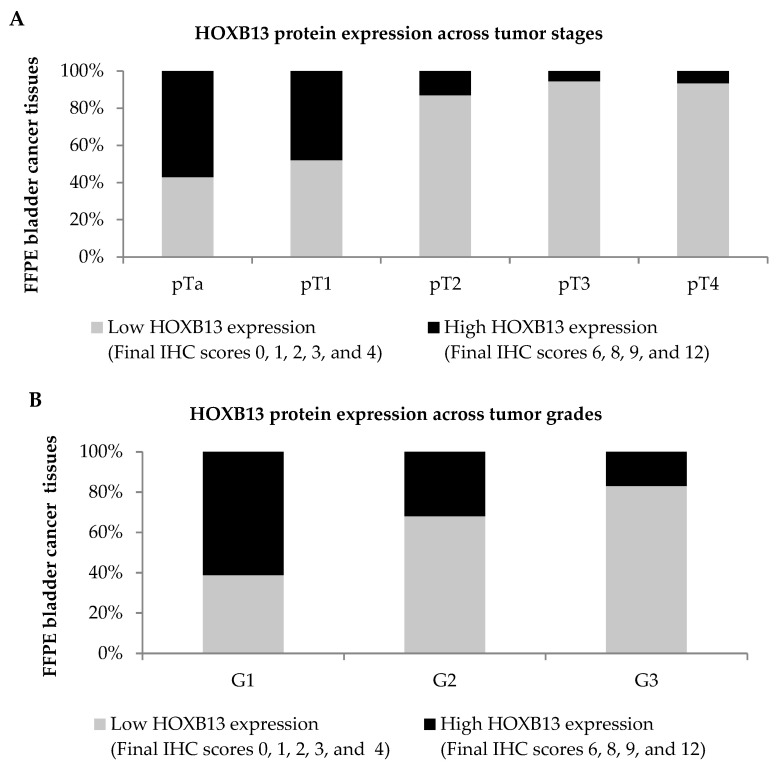
Protein expression of HOXB13 in FFPE bladder cancer tissues. The distribution of HOXB13 protein expression was stratified according to different (**A**) tumor stages and (**B**) tumor grades. HOXB13 protein expression was categorized into low and high HOXB13 expression.

**Table 1 diagnostics-13-02636-t001:** IHC scoring system for HOXA13 protein expression.

Percentage of Positive Cells	Score	Staining Intensity	Score
Less than 1% positive tumor cells	0	No staining	0
1 to 30% positive tumor cells	1	Weak staining	1
30 to 70% positive tumor cells	2	Moderate staining	2
More than 70% positive tumor cells	3	Strong staining	3

Final IHC score = score of percentage of positive cells × score of staining intensity; final IHC score 0, 1, 2, and 3 = low HOXA13 expression; final IHC score 4, 6, and 9 = high HOXA13 expression.

**Table 2 diagnostics-13-02636-t002:** IHC scoring system for HOXB13 protein expression.

Percentage of Positive Cells	Score	Staining Intensity	Score
0% positive tumor cells	0	No staining	0
1 to 10% positive tumor cells	1	Weak staining	1
11 to 50% positive tumor cells	2	Moderate staining	2
50 to 80% positive tumor cells	3	Strong staining	3
More than 80% positive tumor cells	4		

Final IHC score = score of percentage of positive cells × score of staining intensity; final IHC score 0, 1, 2, 3, and 4 = low HOXB13 expression; final IHC score 6, 8, 9, and 12 = high HOXB13 expression.

**Table 3 diagnostics-13-02636-t003:** Association between HOXA13 protein expression and the demographic as well as clinicopathological characteristics of bladder cancer patients.

Characteristic	Number of Cases (%)		HOXA13 Expression				χ^2^ Value	*p* Value	Cramer’s V Value	Strength of Association
			Low ^2^ (%)		High (%)					
Gender							1.265	0.261	-	-
Male	105	(95.5)	37	(35.2)	68	(64.8)				
Female	5	(4.5)	3	(60.0)	2	(40.0)				
Ethnicity							3.132	0.372	-	-
Malay	65	(59.1)	25	(38.5)	40	(61.5)				
Chinese	28	(25.5)	7	(25.0)	21	(75.0)				
Indian	14	(12.7)	6	(42.9)	8	(57.1)				
Others	3	(2.7)	2	(66.7)	1	(33.3)				
Age							1.947	0.163	-	-
<50	13	(11.8)	7	(53.8)	6	(46.2)				
≥50	97	(88.2)	33	(34.0)	64	(66.0)				
Tumor invasiveness							5.384	0.020 *	0.221	Moderate
NMIBC	49	(44.5)	12	(24.5)	37	(75.5)				
MIBC	61	(55.5)	28	(45.9)	33	(54.1)				
Tumor stage							9.350	0.053	-	-
pTa	23	(20.9)	5	(21.7)	18	(78.3)				
pT1	26	(23.6)	7	(26.9)	19	(73.1)				
pT2	23	(20.9)	14	(60.9)	9	(39.1)				
pT3	21	(19.1)	7	(33.3)	14	(66.7)				
pT4	17	(15.5)	7	(41.2)	10	(58.8)				
Tumor grade ^1^							11.414	0.003 *	0.322	Strong
G1	20	(18.2)	3	(15.0)	17	(85.0)				
G2	26	(23.6)	16	(61.5)	10	(38.5)				
G3	64	(58.2)	21	(32.8)	43	(67.2)				
Lymph node metastasis							4.901	0.027 *	0.288	Moderate
Yes	26	(23.6)	9	(34.6)	17	(65.4)				
No	33	(30.0)	21	(63.6)	12	(36.4)				
N/A	51	(46.4)	-	-	-	-				
Metastasis							1.892	0.169	-	-
Yes	30	(27.3)	14	(46.7)	16	(53.3)				
No	80	(72.7)	26	(32.5)	54	(67.5)				
Recurrence							4.092	0.043 *	0.193	Weak
Yes	66	(60.0)	19	(28.8)	47	(71.2)				
No	44	(40.0)	21	(47.7)	23	(52.3)				

Abbreviations: MIBC = muscle invasive bladder cancer; NMIBC = non-muscle invasive bladder cancer; ^1^ WHO 1973 classification system; ^2^ low HOXA13 expression (including those with negative expression); * statistically significant at *p* < 0.05.

**Table 4 diagnostics-13-02636-t004:** Association between HOXA13 protein localization and the demographic as well as clinicopathological characteristics of bladder cancer patients.

Characteristic	Number of Cases (%)		HOXA13 Localization						χ^2^ Value	*p*-Value	Cramer’s V Value	Strength of Association
			None		Nucleus		Nucleus and Cytoplasm					
Gender									0.506	0.777	-	-
Male	105	(95.5)	6	(5.7)	86	(81.9)	13	(12.4)				
Female	5	(4.5)	0	(0.0)	4	(80.0)	1	(20.0)				
Ethnicity									11.728	0.068	-	-
Malay	65	(59.1)	3	(4.6)	53	(81.5)	9	(13.8)				
Chinese	28	(25.5)	0	(0.0)	24	(85.7)	4	(14.3)				
Indian	14	(12.7)	2	(14.3)	12	(85.7)	0	(0.0)				
Others	3	(2.7)	1	(33.3)	1	(33.3)	1	(33.3)				
Age									15.156	0.001 *	0.371	Strong
<50	13	(11.8)	0	(0.0)	7	(53.8)	6	(46.2)				
≥50	97	(88.2)	6	(6.2)	83	(85.6)	8	(8.2)				
Tumor invasiveness									2.068	0.356	-	-
NMIBC	49	(44.5)	1	(2.0)	42	(85.7)	6	(12.2)				
MIBC	61	(55.5)	5	(8.2)	48	(78.7)	8	(13.1)				
Tumor stage									11.257	0.188	-	-
pTa	23	(20.9)	1	(4.3)	20	(87.0)	2	(8.7)				
pT1	26	(23.6)	0	(0.0)	22	(84.6)	4	(15.4)				
pT2	23	(20.9)	4	(17.4)	17	(73.9)	2	(8.7)				
pT3	21	(19.1)	1	(4.8)	18	(85.7)	2	(9.5)				
pT4	17	(15.5)	0	(0.0)	13	(76.5)	4	(23.5)				
Tumor grade ^1^									3.438	0.487	-	-
G1	20	(18.2)	1	(5.0)	15	(75.0)	4	(20.0)				
G2	26	(23.6)	1	(3.8)	20	(76.9)	5	(19.2)				
G3	64	(58.2)	4	(6.2)	55	(85.9)	5	(7.8)				
Lymph node metastasis									1.336	0.513	-	-
Yes	26	(23.6)	1	(3.8)	21	(80.8)	4	(15.4)				
No	33	(30.0)	4	(12.1)	25	(75.8)	4	(12.1)				
N/A	51	(46.4)	-	-	-	-	-	-				
Metastasis									0.140	0.933	-	-
Yes	30	(27.3)	2	(6.7)	24	(80.0)	4	(13.3)				
No	80	(72.7)	4	(5.0)	66	(82.5)	10	(12.5)				
Recurrence									7.566	0.023 *	0.262	Moderate
Yes	66	(60.0)	1	(1.5)	59	(89.4)	6	(9.1)				
No	44	(40.0)	5	(11.4)	31	(70.5)	8	(18.2)				

Abbreviations: MIBC = muscle invasive bladder cancer; NMIBC = non-muscle invasive bladder cancer; ^1^ WHO 1973 classification system; * statistically significant at *p* < 0.05.

**Table 5 diagnostics-13-02636-t005:** Association between HOXB13 protein expression and the demographic as well as clinicopathological characteristics of bladder cancer patients.

Characteristic	Number of Cases (%)		HOXB13 Expression				χ^2^ Value	*p*-Value	Cramer’s V Value	Strength of Association
			Low ^2^ (%)		High (%)					
Gender							0.346	0.556	-	-
Male	97	(95.1)	70	(72.2)	27	(27.8)				
Female	5	(4.9)	3	(60.0)	2	(40.0)				
Ethnicity							3.000	0.392	-	-
Malay	58	(56.9)	43	(74.1)	15	(25.9)				
Chinese	27	(26.5)	18	(66.7)	9	(33.3)				
Indian	14	(13.7)	11	(78.6)	3	(21.4)				
Others	3	(2.9)	1	(33.3)	2	(66.7)				
Age							0.040	0.841	-	-
<50	13	(12.7)	9	(69.2)	4	(30.8)				
≥50	89	(87.3)	64	(71.9)	25	(28.1)				
Tumor invasiveness							23.212	≤0.001 *	0.477	Strong
NMIBC	46	(45.1)	22	(47.8)	24	(52.2)				
MIBC	56	(54.9)	51	(91.1)	5	(8.9)				
Tumor stage							24.010	≤0.001	0.485	Strong
pTa	21	(20.6)	9	(42.9)	12	(57.1)				
pT1	25	(24.5)	13	(52.0)	12	(48.0)				
pT2	23	(22.5)	20	(87.0)	3	(13.0)				
pT3	18	(17.6)	17	(94.4)	1	(5.6)				
pT4	15	(14.7)	14	(93.3)	1	(6.7)				
Tumor grade ^1^							13.427	0.001 *	0.363	Strong
G1	18	(17.6)	7	(38.9)	11	(61.1)				
G2	25	(24.5)	17	(68.0)	8	(32.0)				
G3	59	(57.8)	49	(83.1)	10	(16.9)				
Lymph node metastasis							0.789	0.375	-	-
Yes	24	(23.5)	24	(100.0)	0	(0.0)				
No	31	(30.4)	30	(96.8)	1	(3.2)				
N/A	47	(46.1)	-	-	-	-				
Metastasis							7.976	0.005 *	0.280	Moderate
Yes	27	(26.5)	25	(92.6)	2	(7.4)				
No	75	(73.5)	48	(64.0)	27	(36.0)				
Recurrence							3.410	0.065	-	-
Yes	63	(61.8)	41	(65.1)	22	(34.9)				
No	39	(38.2)	32	(82.1)	7	(17.9)				

Abbreviations: MIBC = muscle invasive bladder cancer; NMIBC = non-muscle invasive bladder cancer; ^1^ WHO 1973 classification system; ^2^ low HOXB13 expression (including those with negative expression); * statistically significant at *p* < 0.05.

**Table 6 diagnostics-13-02636-t006:** Association between HOXB13 protein localization and the demographic as well as clinicopathological characteristics of bladder cancer patients.

Characteristic	Number of Cases (%)		HOXB13 Localization								χ^2^ Value	*p*-Value	Cramer’s V Value	Strength of Association
			None		Nucleus		Nucleus and Cytoplasm		Cytoplasm					
Gender											1.437	0.697	-	-
Male	97	(95.1)	41	(42.3)	34	(35.1)	19	(19.6)	3	(3.1)				
Female	5	(4.9)	2	(40.0)	1	(20.0)	2	(40.0)	0	(0.0)				
Ethnicity											2.216	0.988	-	-
Malay	58	(56.9)	25	(43.1)	18	(31.0)	13	(22.4)	2	(3.4)				
Chinese	27	(26.5)	10	(37.0)	11	(40.7)	5	(18.5)	1	(3.7)				
Indian	14	(13.7)	7	(50.0)	5	(35.7)	2	(14.3)	0	(0.0)				
Others	3	(2.9)	1	(33.3)	1	(33.3)	1	(33.3)	0	(0.0)				
Age											3.346	0.341	-	-
<50	13	(12.7)	5	(38.5)	3	(23.1)	5	(38.5)	0	(0.0)				
≥50	89	(87.3)	38	(42.7)	32	(36.0)	16	(18.0)	3	(3.4)				
Tumor invasiveness											26.927	≤0.001 *	0.514	Strong
NMIBC	46	(45.1)	8	(17.4)	27	(58.7)	10	(21.7)	1	(2.2)				
MIBC	56	(54.9)	35	(62.5)	8	(14.3)	11	(19.6)	2	(3.6)				
Tumor stage											43.160	≤0.001 *	0.376	Strong
pTa	21	(20.6)	2	(9.5)	16	(76.2)	2	(9.5)	1	(4.8)				
pT1	25	(24.5)	6	(24.0)	11	(44.0)	8	(32.0)	0	(0.0)				
pT2	23	(22.5)	13	(56.5)	3	(13.0)	7	(30.4)	0	(0.0)				
pT3	18	(17.6)	12	(66.7)	3	(16.7)	1	(5.6)	2	(11.1)				
pT4	15	(14.7)	10	(66.7)	2	(13.3)	3	(20.0)	0	(0.0)				
Tumor grade ^1^											13.769	0.032 *	0.260	Moderate
G1	18	(17.6)	1	(5.6)	10	(55.6)	6	(33.3)	1	(5.6)				
G2	25	(24.5)	13	(52.0)	9	(36.0)	3	(12.0)	0	(0.0)				
G3	59	(57.8)	29	(49.2)	16	(27.1)	12	(20.3)	2	(3.4)				
Lymph node metastasis											0.154	0.985	-	-
Yes	24	(23.5)	17	(70.8)	3	(12.5)	3	(12.5)	1	(4.2)				
No	31	(30.4)	23	(74.2)	4	(12.9)	3	(9.7)	1	(3.2)				
N/A	47	(46.1)	-	-	-	-	-	-	-	-				
Metastasis											16.309	0.001 *	0.400	Strong
Yes	27	(26.5)	20	(74.1)	3	(11.1)	3	(11.1)	1	(3.7)				
No	75	(73.5)	23	(30.7)	32	(42.7)	18	(24.0)	2	(2.7)				
Recurrence											5.611	0.132	-	-
Yes	63	(61.8)	21	(33.3)	24	(38.1)	16	(25.4)	2	(3.2)				
No	39	(38.2)	22	(56.4)	11	(28.2)	5	(12.8)	1	(2.6)				

Abbreviations: MIBC = muscle invasive bladder cancer; NMIBC = non-muscle invasive bladder cancer; ^1^ WHO 1973 classification system; * statistically significant at *p* < 0.05.

## Data Availability

The data presented in this study are available on request from the corresponding author.

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
