# Peer review of "Differential Protein Expression Patterns of HOXA13 and HOXB13 Are Associated with Bladder Cancer Progression"

_diagnostics, 2023, doi:10.3390/diagnostics13162636_

Round 1
Reviewer 1 Report
The manuscript entitled "Differential Protein Expression Patterns of HOXA13 and HOXB13 Are Associated with Bladder Cancer" Progressionpresents very interesting results. The work is prepared in a clear and logical way. However, as it concerns tumor markers and the relationship with prognosis, the authors should prepare it in accordance with the REportnig recommendations for tumour MARKer prognostic studies (REMARK) guidelines. I recommend adapting the paper to this guidelines, especially since the authors do not provide any limitations to their analyzes while the reference group is not ideally selected. It consisted of samples of other normal tissue and cancer tissue: "FFPE human skin tissue was used as the positive control for HOXA13 IHC analysis, while FFPE human prostate cancer tissue was used as the positive control for HOXB13 IHC analysis". The number of samples qualified to each of the control groups is also not specified. There is also no information on sensitivity, specificity, positive predictive value and negative predictive value. The tests were performed in the tissue and therefore it is necessary to undergo surgery to perform the tests.
Round 2
Reviewer 1 Report
I consider the authors' explanations and the changes introduced to the paper to be sufficient.